# Rethinking IL-1 Antagonism in Respiratory Viral Infections: A Role for IL-1 Signaling in the Development of Antiviral T Cell Immunity

**DOI:** 10.3390/ijms242115770

**Published:** 2023-10-30

**Authors:** Bram Van Den Eeckhout, Marlies Ballegeer, Jozefien De Clercq, Elianne Burg, Xavier Saelens, Linos Vandekerckhove, Sarah Gerlo

**Affiliations:** 1HIV Cure and Research Center (HCRC), 9000 Ghent, Belgiumjozefien.declercq@ugent.be (J.D.C.);; 2Department of Internal Medicine and Pediatrics, Ghent University, 9000 Ghent, Belgium; 3Department of Biomolecular Medicine, Ghent University, 9820 Ghent, Belgium; 4VIB Center for Medical Biotechnology, 9052 Ghent, Belgium; marlies.ballegeer@vib-ugent.be (M.B.);; 5Department of Biochemistry and Microbiology, Ghent University, 9000 Ghent, Belgium

**Keywords:** interleukin-1, CD8 T cells, influenza A virus, infection

## Abstract

IL-1R integrates signals from IL-1α and IL-1β, and it is widely expressed across tissues and immune cell types. While the expression pattern and function of IL-1R within the innate immune system is well studied, its role in adaptive immunity, particularly within the CD8 T cell compartment, remains underexplored. Here, we show that CD8 T cells dynamically upregulate IL-1R1 levels during priming by APCs, which correlates with their proliferation status and the acquisition of an effector phenotype. Notably, this IL-1 sensitivity persists in memory CD8 T cells of both mice and humans, influencing effector cytokine production upon TCR reactivation. Furthermore, our study highlights that antiviral effector and tissue-resident CD8 T cell responses against influenza A virus infection become impaired in the absence of IL-1 signaling. Altogether, these data support the exploitation of IL-1 activity in the context of T cell vaccination strategies and warrant consideration of the impact of clinical IL-1 inhibition on the rollout of T cell immunity.

## 1. Introduction

Expression of the interleukin-1 (IL-1) receptor (IL-1R) complex on CD4 T cell subsets has been extensively documented [1,2]. Functionally, IL-1 has mainly been described as a polarizing cytokine that facilitates the differentiation of naive CD4 T cells towards the Th17 cell phenotype [3,4,5], and as a driver of CD4 T cell proliferation, effector functionality, and memory differentiation in response to cognate antigen recognition [6,7,8]. In contrast to CD4 T cells, knowledge on the expression pattern and functional importance of the IL-1R complex in CD8 T cells is more limited. Nevertheless, we [9,10] and others [11,12,13] have shown that IL-1 can also modulate the functions of CD8 T cells.

In the clinic, therapeutic inhibition of IL-1 signaling has been strategically used to relieve the hyperinflammatory state in patients who suffer from life-threatening cytokine storm syndrome after SARS-CoV-2 infection. While treatment with the IL-1R antagonist (IL-1Ra) anakinra improved patient survival and shortened hospitalization [14], no significant beneficial effect on survival could be observed in patients who received the IL-1β-neutralizing monoclonal antibody canakinumab [15], which demonstrates the variable success of this intervention. In a Cochrane meta-analysis, Davidson and colleagues were unable to evidence a beneficial role of IL-1 blocking agents in the treatment of COVID-19 [16].

In this Brief Report, we demonstrate that CD8 T cells dynamically upregulate their responsiveness to IL-1 during priming by antigen-presenting cells, and that memory CD8 T cells can maintain elevated IL-1R1 levels. We further show that the rollout of antiviral CD8 T cell immunity is curtailed in the absence of IL-1 signaling during respiratory virus infection. While IL-1 antagonism can undoubtedly be of high value for selected groups of patients with severe virus-induced pneumonia, our data underline a beneficial and supporting role for this pro-inflammatory cytokine in the formation of functional and durable T cell immune responses to viral infections. We believe that this information should be considered when strategically deploying IL-1 antagonism for the treatment of viral-infection-associated hyperinflammation. Furthermore, our data show that IL-1 activity might be desired in vaccination strategies or immunotherapies that are aimed at mounting strong and durable T cell immune responses.

## 2. Results

### 2.1. CD8 T Cells Dynamically Modulate IL-1R1 Levels during Priming and Maintain Elevated Sensitivity to IL-1 as Memory Cells

Beyond the work of Kothari and colleagues [1], who showed high IL-1 sensitivity in human CD8 T cells that express CD161 (identified as circulating mucosal-associated invariant T (MAIT)-like cells), data on IL-1R1 expression in the CD8 T cell subset are scarce. Therefore, we further characterized the IL-1R1 expression pattern in CD8 T cells. We started by studying IL-1R1 expression levels in CD8 T cells from mice during priming by antigen-presenting cells (APCs). To do so, we established co-cultures of SIINFEKL peptide-loaded bone-marrow-derived dendritic cells (BM-DCs) and CD8 T cells with a transgenic T cell receptor (TCR) that binds the SIINFEKL epitope of ovalbumin (OVA_257-264_) (from here on referred to as OT-I CD8 T cells), and we used flow cytometry to measure IL-1R1 levels. Following priming, OT-I CD8 T cells initiated proliferation and acquired an effector T cell (Teff cell) phenotype that was characterized by enhanced expression of the lineage transcription factor T-bet and the integrin CD44. We found that these activated OT-I CD8 Teff cells upregulated the IL-1R1 on their cell membrane, which was positively correlated with their proliferation index (i.e., their stage of cell division) (Figure 1A,B, see Appendix A for the gating strategy). Furthermore, we validated this observation in vivo by performing immunization of naive C57BL/6 mice with SIINFEKL-loaded BM-DCs. One week after transfer of these BM-DCs, higher levels of IL-1R1 could be observed on splenic CD8 T cells that produced the effector cytokine IFN-γ after ex vivo restimulation with SIINFEKL compared with their counterparts, wherein no IFN-γ accumulation was measured (Figure 1C, see Appendix A for the gating strategy). Together, these data show that CD8 T cells enhance their sensitivity to IL-1 during priming by dynamic upregulation of IL-1R1 expression levels, which correlates with the acquisition of a Teff cell phenotype.

Next, we evaluated whether a differential expression pattern of IL-1R1 could be observed between naive and antigen-experienced T cell subsets in both mice and humans. Flow cytometric analysis of IL-1R1 expression levels on splenic murine CD4 and CD8 T cell subsets revealed higher cytokine receptor levels on CD44^+^CD62L^−^ antigen-experienced effector memory T cells (Tem cells) compared with CD44^−^CD62L^+^ naive T cells (Tn cells). Consistently higher surface levels of IL-1R1 were observed on mouse CD8 T cells compared with CD4 T cells (Figure 1D; see Appendix A for the gating strategy). To assess the human situation, we employed intracellular flow cytometry to measure phospho-NF-κB (p-NF-κB) following in vitro treatment of peripheral blood mononuclear cell (PBMC) cultures with IL-1β [17]. This indirect demonstration of IL1-1R1 expression was performed because no specific anti-human-IL1-R1 antibody was available. Data from multiple donors revealed higher levels of p-NF-κB in memory (CD45RO^+^CD45RA^−^) CD8 T cells compared with naive (CD45RO^−^CD45RA^+^) CD8 T cells (Figure 1E; see Appendix A for the gating strategy). These data suggest that the higher sensitivity to IL-1 cytokines acquired by CD8 Teff cells during the priming phase becomes hardwired when these cells progress to form memory subsets.

### 2.2. Neutrophils Are the Predominant Source of IL-1β in the Lungs during Influenza A Virus Infection

IL-1 superfamily cytokines are strongly recognized as damage-associated molecular patterns (DAMPs) that are released in response to pathogen detection or (non-)sterile disruption of tissue homeostasis, and they are pivotal in driving peripheral inflammation [2]. Steuerman and colleagues have demonstrated *Il1b* expression in granulocytes retrieved from the lung parenchyma of C57BL/6 mice during infection with influenza A virus (IAV) strain PR8 (H1N1), and they indicated that *Il1b* is one of the earliest genes that is expressed during the granulocytes’ antiviral trajectory [18]. To confirm these findings experimentally, we infected C57BL/6 mice intranasally with PR8 IAV and interrogated the myeloid compartment in infected lungs 48 h post-inoculum. The morbidity of the infection was illustrated by a loss in body weight over time (Appendix A). FACS analysis of the total leukocyte population (all CD45^+^ cells) revealed that the myeloid compartment in IAV-infected lung parenchyma is dominated by neutrophils (mean = 31.88%; SEM = 4.38%) (Figure 2A–C). MHC-II^low^ macrophages (mean = 13.6%; SEM = 1.86%), monocytes (mean = 5.22%; SEM = 0.37%), cDC2s (mean = 3.4%; SEM = 0.55%), and MHC-II^high^ macrophages (mean = 3.27%; SEM = 0.33%) make up much smaller fractions of the myeloid repertoire. It should be noted that we did not detect cDC1s at this timepoint, which could be a consequence of their migration towards lymphoid tissue [19]. The release of matured and bioactive IL-1β could only be detected in conditioned supernatants of freshly isolated neutrophils after 24 h of in vitro culture (Figure 2D). These data show that IAV creates environments that are enriched in IL-1 cytokines over the course of pulmonary infection, and we can indicate neutrophils as the predominant source of IL-1β in the lungs during infection with IAV.

### 2.3. Loss of IL-1R1 Signaling during IAV Infection Is Associated with Impaired Formation and Functionality of Antiviral CD8 Teff and CD8 Trm Cells

Thus far, we have demonstrated that CD8 Teff cells display elevated IL-1 sensitivity and enter IAV-infected tissues that are enriched in neutrophil-derived pro-inflammatory IL-1β. Next, we asked whether the formation of virus-specific CD8 T cell immunity is impaired in the absence of IL-1 signaling. To do so, we infected wild-type (WT) and *Il1r1*^−/−^ (KO) C57BL/6 mice intranasally with a sublethal dose of PR8 IAV and compared the formation of antiviral CD8 Teff cells seven days post-inoculation. Fewer IAV-specific CD8 Teff cells could be retrieved from the lung parenchyma of infected KO mice compared with WTs (although this was not statistically significant, *p* = 0.09) (Figure 3A; see Appendix A for the gating strategy). In addition, IAV-specific CD8 Teff cells from KO mice were also poised more towards a naive-like rather than an effector-like phenotype, based on CD44/CD62L surface expression (Figure 3B), and showed lower intracellular levels of the transcription factors T-bet and Eomes (Figure 3C). This finding can potentially be explained by the observation that IL-1R1 signaling, in T cells from wild-type mice, leads to the phosphorylation and activation of the transcription factor NF-κB [2], which binds to the *Eomes* promotor in Teff cells [20]. Furthermore, it has been demonstrated that p-NF-κB-induced Eomes regulates the expression of the anti-apoptotic protein Bcl-2 [21]. Building further on these observations, we measured reduced amounts of Bcl-2 in IAV-specific CD8 Teff cells in the infected lung parenchyma of KO mice (Figure 3D). Comparable results could be obtained in IAV-specific CD8 Teff cells retrieved inside the pulmonary vasculature (Appendix A).

Given the central role of Bcl-2 as a pro-survival factor during the establishment of stable memory T cell pools following pathogen clearance [21], we examined whether the formation and functionality of antiviral tissue-resident memory CD8 T cells (CD8 Trm cells) was affected in infected KO mice. Six weeks after inoculation of PR8 H1N1 IAV, we retrieved fewer virus-specific CD8 Trm cells from the lungs of KO animals compared with WTs (although this was not statistically significant, *p* = 0.0792) (Figure 3E; see Appendix A for the gating strategy). Upon restimulation of these cells ex vivo with pools of PR8 H1N1 IAV nucleoprotein (NP)-derived peptides (ASNENMETM, RLIQNSLTI and GERQNATEI, all H2-Db-restricted), we measured lower frequencies of polyfunctional memory CD8 T cells that released both of the effector cytokines IFN-γ and TNF in the lungs (not statistically significant, *p* = 0.1048) and spleens of KO mice (Figure 3F; see Appendix A for the gating strategy). At this memory timepoint, we also started to treat mice with FTY720, a pharmacological inhibitor of the sphingosine-1-phosphate receptor (S1PR) that antagonizes the ability of lymphocytes to egress from lymphatic tissues [22]. Infection of FTY720-treated mice with a lethal dose of X31 (H3N2) IAV thus allowed us to compare the heterosubtypic protective capacity of tissue-resident memory cells between KO and WT animals. However, despite the immunological impact of IL-1R1 signaling on the amounts and effector functionality of IAV-specific CD8 Trm cells, we could not observe statistically significant differences in survival between KO and WT mice (death in 3/10 KOs compared with 1/8 WTs, *p* = 0.3997) (Appendix A).

To translate the findings from Figure 3F to the human setting, purified CD8 T cells obtained from clinically confirmed *Cytomegalovirus* (CMV)-positive donors were stimulated with IL-1β or left unstimulated. Following stimulation, the CD8 T cells were extensively washed and co-cultured with autologous monocyte-derived DCs (moDCs) loaded with CMV peptide pools. We observed more IFN-γ production in CD8 Tem cells that had been pretreated with IL-1β compared with cells that had not been preconditioned (Figure 3G; see Appendix A for the gating strategy). Interestingly, stimulation of cells with IL-1β did not have an effect on the activation of CD8 Tn cells.

## 3. Discussion

IL-1R is a heterodimer composed of the primary IL-1R1 and the accessory IL-1R3 (or IL-1RAcP) chain. This complex integrates signals delivered by the cytokines IL-1α and IL-1β, and it displays a highly pleiotropic expression pattern across tissues and (immune) cell types. IL-1 cytokines are typical pro-inflammatory mediators that act as endogenous “danger” signals in the body, and as alarm cells of the innate immune system in response to pathogenic insult or sterile injury [2]. While the importance and role of the IL-1R complex within innate immune cells, such as macrophages and neutrophils, are well appreciated [23], few studies have reported on the expression pattern and functions of this receptor in cells of the adaptive immune system, and the CD8 T cell subset remains especially understudied [2]. Such information is important, because activated CD8 T cells infiltrate inflamed tissues that are enriched in IL-1 cytokines during bacterial or viral infections, and such pro-inflammatory microenvironments might influence the phenotype, function, and fate of the entering effector cells. Therefore, a better understanding of how T cells operate in these “heated” environments might allow us to more closely emulate the natural T cell response to infection in the context of prophylactic vaccination, as well as allowing us to improve T cell-based vaccination strategies. In turn, this information could help to better comprehend the impact of therapeutic inhibition of IL-1 signaling on the rollout of antiviral CD8 T cell immunity.

A key novel finding of our study is that CD8 T cells dynamically increase surface IL-1R1 levels during priming by APCs, and this IL-1R1 upregulation appears to correlate with the proliferation index and effector status of newly activated CD8 T cells. We hypothesize that, in this way, newly primed and activated CD8 Teff cells are prepared to operate better in environments that are enriched in IL-1 cytokines during active pathogen infection. Such a mechanism might also be in play beyond the context of CD8 T cell priming, as a recent work by Kim et al. demonstrated that IL-1R1 expression can be increased in activated human CD8 T cells via CD4 T cells that secrete IL-21 [24].

Importantly, our data show that the further rollout of antiviral CD8 T cell immunity is curtailed in the absence of IL-1 signaling during infection. In accordance with several earlier studies [19,25,26], we observed a reduction in the amount of virus-specific CD8 Teff cells that had infiltrated the lung parenchyma of full-body *Il1r1^−/−^* KO C57BL/6 mice compared with WT animals following infection with IAV. Fewer of these antiviral CD8 Teff cells adopted a typical effector phenotype, based on expression of the surface integrins CD44/CD62L and the transcription factor T-bet. In addition, we observed decreased levels of the transcription factor Eomes [27,28] and the pro-survival factor Bcl2 [29] in antiviral CD8 Teff cells from KO animals, which suggests a diminished potential of these cells to persist in the periphery and establish a pool of tissue-resident memory CD8 T cells (CD8 Trm cells). Indeed, 6 weeks after PR8 IAV infection, we found fewer antiviral CD8 Trm cells in lungs of KO animals compared to WTs. This finding corresponds well to earlier data from our group [9] and others [30,31], where the inclusion of IL-1 activity in prophylactic vaccine formulations against IAV or respiratory syncytial virus augmented the formation of CD8 Trm cells in the lungs. Inversely, patients on anakinra therapy are more susceptible to influenza, pneumococci, human papillomavirus, and herpes zoster infections [32]. In addition, some authors have suggested delaying COVID-19 vaccination in patients on long-acting inhibitors (e.g., canakinumab and rilonacept) until their drug levels return to baseline in order to obtain the optimal vaccine responses [33]. Such a delay might, however, depend on the molecular nature of the vaccine, as influenza and meningococcal vaccines have proven to be effective in healthy human volunteers who were treated with canakinumab [34]. A limitation of our experimental model is the use of full-body *Il1r1^−/−^* animals, from which we cannot deduct whether the experimental observations following infection were necessarily driven by direct IL-1 activity on CD8 T cells.

Furthermore, we found that both mouse and human CD8 T cells can retain elevated sensitivity to IL-1 cytokines when acquiring a memory phenotype. While ours is the first study to report this observation in memory CD8 T cells, Jain et al. and Kothari et al. have demonstrated this previously in murine [8] and human [1] CD4 Tem cells, which implies that IL-1 sensitivity is likely regulated in a similar manner in both T cell subsets. Functionally, Jain et al. demonstrated that IL-1R activation in memory CD4 T cells is essential to enable lineage cytokine production in Th1, Th2, and Th17 cells following TCR ligation [8], and from our data we may attribute a comparable role to the IL-1R complex in memory CD8 T cells. Following ex vivo TCR reactivation with IAV peptides, reduced production of the effector cytokines IFN-γ and TNF could be observed in memory CD8 T cells from KO mice compared with WT animals. In a human context, IL-1β pre-treatment of CD8 T cells derived from CMV-seropositive individuals resulted in increased IFN-γ production after co-culture with autologous moDCs that presented CMV-derived peptide pools. A potential mechanism that could underlie these observations might be the stabilization of cytokine mRNA transcripts after IL-1R complex activation [35]. This has been experimentally demonstrated to occur via a p38 MAPK-dependent mechanism for *Ifng* mRNA stabilization in murine memory CD4 T cells [8]. This stabilization requires the presence of an AU-rich motif in the 3′ UTR, which is also present in the mouse *Tnfa* mRNA [35].

In conclusion, we demonstrated that infection with IAV leads to the establishment of a pro-inflammatory milieu in the lungs via the infiltration of neutrophils that secrete mature and bioactive IL-1β. While neutrophils are clearly the main source of IL-1β within the myeloid compartment early in the infection (i.e., 48 h post-inoculation), other cells might take over this role when the amount of infiltrating neutrophils starts to wane over time [36]. The absence of IL-1R signaling during viral infection results in reduced antiviral effector and tissue-resident memory CD8 T cell immunity. We showed that CD8 T cells prepare themselves to operate in inflamed environments via the upregulation of surface IL-1R1 levels during priming by APCs, and we demonstrated that CD8 T cells can retain this increased sensitivity to IL-1 cytokines when acquiring a memory phenotype. These findings should be carefully considered during the therapeutic inhibition of IL-1 in the context of respiratory viral infections, and they might be taken into consideration in the design of next-generation T cell vaccines.

## 4. Materials and Methods

### 4.1. Mice

Female C57BL/6 mice were purchased from Charles River Laboratories. OT-I TCR transgenic CD45.1 *Rag2*^−/−^ C57BL/6 mice (Bart Lambrecht, VIB-UGent) and full-body *Il1r1*^−/−^ C57BL/6 mice (Andy Wullaert, VIB-UGent) were bred and crossed in our animal facilities. As a consequence of the *Rag2*^−/−^, OT-I mice have no functional T and B cells apart from the T cells that carry the transgenic TCR. Full-body *Il1r1* knockout causes defective responses to certain inflammatory agents and altered responses to various pathogenic organisms. All of the animals were housed under pathogen-free conditions in IVCs and temperature-controlled environments with a 12 h day/12 h night cycle. The mice received water and food ad libitum. The FELASA guidelines were followed in all experiments, and approval was obtained from the Ethical Committee of the Faculty of Medicine and Health Sciences (UGent) (ECD21-12K, ECD21-13, and ECD2021-031). The mice were randomly allocated to treatment groups, and the investigators were not blinded during data collection and analysis.

### 4.2. In Vitro OT-I CD8 Co-Culture Experiments

Bone marrow was isolated from the tibias and femurs of C57BL/6 mice by flushing the bones with PBS (14190-169, Thermo Fisher Scientific, Waltham, MA, USA). Red blood cells (RBCs) were lysed with ACK lysis buffer (A1049201, Thermo Fisher Scientific, Waltham, MA, USA), and the cells were seeded in 6-well plates in RPMI-1640 (61870-044, Thermo Fisher Scientific, Waltham, MA, USA) + 10% FBS + penicillin/streptomycin (100×) (15140122, Thermo Fisher Scientific, Waltham, MA, USA) + β-mercaptoethanol (500×) (21985-023, Thermo Fisher Scientific, Waltham, MA, USA) + recombinant mouse GM-CSF (20 ng/mL) (576302, BioLegend, San Diego, CA, USA). After 10 days of culture, mature BM-DCs (2.5 × 10^6^ cells/mL) were pulsed with 10 pM–1 nM SIINFEKL (OVA_257-264_) (AS-60193-1, Anaspec, Freemont, CA, USA) for 2 h at 37 °C, 5% CO_2_. Spleens were isolated from OT-I TCR transgenic CD45.1 *Rag2*^−/−^ C57BL/6 mice and processed to single cells. CD8 T cells were purified by negative selection using magnetic-activated cell sorting (MACS) (130-104-075, Miltenyi Biotech, Bergisch Gladbach, Germany), following the manufacturer’s instructions. Cells were subsequently labeled with 5 μM of CellTrace Violet (CTV) (C34557, Thermo Fisher Scientific, Waltham, MA, USA) and plated (10^5^ cells/well) together with the loaded BM-DCs (10^4^ cells/well) in 96-well plates. After 72 h, the cells were blocked for 30 min at 4 °C with anti-CD16/32 (100×) (clone 93, 14-0161-82, Thermo Fisher Scientific, Waltham, MA, USA). Sample fixation and permeabilization were performed using the Foxp3 Transcription Factor Staining Buffer Set (00-5523-00, Thermo Fisher Scientific, Waltham, MA, USA). The cells were stained for 1 h at 4 °C with anti-CD45.1 APC-Cy7 (500×) (clone A20, 560579, BD Biosciences, Erembodegem, Belgium), T-bet PE-Cy7 (100×) (clone 4B10, 644823, BioLegend, San Diego, CA, USA), CD44 BV711 (100×) (clone IM7, 103057, BioLegend, San Diego, CA, USA) and IL-1R1 biotin (100×) (clone JAMA-147, 113503, BioLegend, San Diego, CA, USA). Secondary staining was performed for 1 h at 4 °C with anti-biotin streptavidin APCs (100×) (405207, BioLegend, San Diego, CA, USA). The samples were recorded on a three-laser BD LSRFortessa flow cytometer (BD Biosciences, Erembodegem, Belgium). Data were analyzed using FlowJo software (version 10.8.1) (Treestar, Ashland, OR, USA).

### 4.3. Immunization with DC-SIINFEKL

BM-DCs pulsed with SIINFEKL were generated as described above and transferred i.v. (3.0 × 10^6^ cells in 200 µL PBS) in naive C57BL/6 mice. Seven days after immunization, the mice were euthanatized, and splenocytes were processed to single cells as described above and stimulated in RPMI-1640 + 10% FBS + penicillin/streptomycin for 5 h at 37 °C and 5% CO_2_ with SIINFEKL (100 nM) in the presence of brefeldin A (1000×) (00-4506-51, Thermo Fisher Scientific, Waltham, MA, USA). Fc blocking, cell fixation, and permeabilization were performed as described above. Cells were stained for 1 h at 4 °C with LIVE/DEAD Fixable Aqua viability dye (1000×) (L34957, Thermo Fisher Scientific, Waltham, MA, USA), anti-CD3 FITC (250×) (clone 17A2, 555274, BD Biosciences, Erembodegem, Belgium), CD8 PE (250×) (clone 53-6.7, 553035, BD Biosciences, Erembodegem, Belgium), IFN-γ APCs (100×) (clone XMG1.2, 506303, BioLegend, San Diego, CA, USA), and IL-1R1 biotin. Secondary staining was performed for 1 h at 4 °C with anti-biotin streptavidin BV711 (100×) (405241, BioLegend, San Diego, CA, USA). The samples were recorded on a four-laser Attune Nxt flow cytometer (Thermo Fisher Scientific, Waltham, MA, USA). Data were analyzed using FlowJo software.

### 4.4. Detection of IL-1R1 Surface Expression Levels on Murine Splenocytes

Spleens were isolated from C57BL/6 mice and processed to single cells. RBCs were lysed as described above. Single cells were blocked (30 min, 4 °C) with anti-mouse CD16/32 and stained for 1 h at 4 °C with anti-mouse CD3 PE-Cy7 (250×) (clone 145-2C11, 552774, BD Biosciences, Erembodegem, Belgium), CD4 FITC (clone RM4-5, 100510, BioLegend, San Diego, CA, USA), CD8 PE, CD44 PerCP-Cy5.5 (100×) (clone IM7, 103032, BioLegend, San Diego, CA, USA), CD62L APC-Cy7 (100×) (clone MEL-14, 104428, BioLegend, San Diego, CA, USA), and IL-1R1 biotin. Secondary staining for 1 h at 4 °C was performed with anti-biotin streptavidin APCs. Samples were recorded on a four-laser Attune Nxt flow cytometer (Thermo Fisher Scientific, Waltham, MA, USA). Data were analyzed using FlowJo software.

### 4.5. Human PBMC Isolation

PBMCs were isolated from whole blood from healthy donors (VIM participants) or HIV-positive individuals on antiretroviral therapy (ART) (ACS participants). Separation was performed by placing the whole blood in a Leucosep™ tube (227290, Greiner Bio-One, Vilvoorde, Belgium) prefilled with 15 mL Lymphoprep™ (07851, STEMCELL Technologies, Saint Égrève, France), followed by a centrifugation step of 30 min at 400× *g* (acceleration 5, brake 3). After isolation, the PBMCs were washed in PBS pH 7.2 (20012027, Thermo Fisher Scientific, Waltham, MA, USA) and centrifuged at 350× *g* for 10 min. The isolated PBMCs were counted, cryopreserved in 1 mL of FCS + 10% DMSO, and stored in liquid nitrogen.

### 4.6. Assessment of NF-κB Phosphorylation by Flow Cytometry on Human PBMCs

Our phosflow protocol for NF-κB activation was based on a previously published protocol [17]. PBMCs were thawed in pre-warmed complete RPMI-1640. After centrifugation, the PBMCs (2 × 10^6^/mL) were seeded in a 25 cm^2^ cell culture flask and rested overnight. Twenty-four hours later, the PBMCs were centrifuged (350× *g*), resuspended in pre-warmed PBS + 2% human serum (092938249, MP Biomedicals), and seeded at 10^6^ cells/well in a 96-well V-bottomed plate. After centrifugation (350× *g*), the cells were resuspended in 50 µL of PBS + 2% human serum with Fixable Viability Dye eFluor™ 780 (65-0865-14, BD Biosciences, Erembodegem, Belgium) and incubated for 5 min at 37 °C. The cells were stimulated with human IL-1β (0.5 nM) (recombinantly produced and purified as described in [9]) and incubated for 15 min at 37 °C. After stimulation, the cells were washed with 100 µL of PBS + 2% human serum and fixed by adding 100 µL of pre-warmed PBS pH 7.2 and 100 µL of pre-warmed CytofixTM Fixation Buffer (554655, BD Biosciences, Erembodegem, Belgium). After 10 min of incubation at 37 °C, the cells were centrifuged for 4 min (600× *g*), washed with 200 µL of PBS + 2% human serum, and stained with anti-human CD45RO BV421 (clone UCHL1, 555493, BD Biosciences, Erembodegem, Belgium), CD3 BV510 (clone UCHT1, 563109, BD Biosciences, Erembodegem, Belgium), CD4 BV605 (clone L200, 562843, BD Biosciences, Erembodegem, Belgium), CD27 BB700 (clone M-T271, 566450, BD Biosciences, Erembodegem, Belgium), and CD8 AF700 (clone RPA-T8, 56-0088-42, Thermo Fisher Scientific, Waltham, USA) in a total volume of 50 µL for 45 min. After washing, the plate was vortexed and the cells were permeabilized by adding 100 µL of pre-cooled BD Phosflow™ Perm Buffer III (558050, BD Biosciences, Erembodegem, Belgium), followed by incubation on ice for 20 min. The cells were washed with 100 µL PBS + 10% human serum, centrifuged (600× *g* at 4 °C), and rehydrated with 200 µL of PBS + 10% human serum. After 15 min of incubation on ice, the plate was centrifuged (600× *g* at 4 °C) for 10 min and stained with anti-pS529 p65 NF-κB PE (clone K10-895.12.50, 558423, BD Biosciences, Erembodegem, Belgium) and anti-CD45RA APC (clone HI100, 304112, BioLegend, San Diego, USA) in a staining volume of 50 µL for 30 min. The PBMCs were washed, resuspended in 100 µL of PBS, and recorded on a three-laser BD LSRFortessa flow cytometer (BD Biosciences, Erembodegem, Belgium). Data were analyzed using FlowJo software.

### 4.7. Influenza Viruses and Viral Infection Procedures

PR8 H1N1 IAV (A/Puerto Rico/8/1934) and X31 H3N2 IAV (A/Aichi/2/68) were grown on Madin–Darby canine kidney (MDCK) cells in serum-free DMEM complemented with TPCK-treated trypsin (T1426, Sigma-Aldrich, Burlington, VT, USA). The mice were anesthetized with a mixture of ketamine (10 mg/kg) and xylazine (60 mg/kg) and challenged by intranasal administration of 50 μL of virus diluted in phosphate-buffered saline (PBS). Body weight was measured daily for 14 days, and mice that had lost 25% or more of their initial body weight were euthanatized. All influenza virus infections were conducted in a BSL2-accredited animal facility. In the reinfection experiment, the mice were treated intraperitoneally with 1 mg/kg FTY720 (SML0700-5MG, Sigma-Aldrich, Burlington, VT, USA) every 24 h. FTY720 treatment began 3 days prior to and was maintained throughout the infection.

### 4.8. Analyses of Myeloid Cells and T Cells in the Lungs upon Influenza Virus Infection

Infected mice were euthanatized by intraperitoneal injection of an overdose of ketamine (80 mg/kg) (Eurovet) and xylazine (5 mg/kg) (Bayer) in 500 µL of PBS. Their lungs were perfused with PBS supplemented with heparin (50 IU/mL) (H5515-100KU, Sigma-Aldrich, Burlington, VT, USA) and collected in PBS on ice. The lungs were chopped into small pieces using scissors and digested for 30 min at 37 °C in RPMI-1640 + HEPES (11300902, Thermo Fisher Scientific, Waltham, MA, USA) supplemented with collagenase D (2.5 mg/mL) (11088858001, Sigma-Aldrich, Burlington, VT, USA) and DnaseI (5 IU/mL) (EN0521, Thermo Fisher Scientific, Waltham, MA, USA). Enzyme-digested lung pieces and complete spleens were minced mechanically on 50 μm cell strainers, after which the RBCs were lysed. For FACS of myeloid cell populations from infected lungs at a hyperacute (48 h) timepoint, Fc blocking was performed as described above, and the cells were stained for 1 h at 4 °C with LIVE/DEAD Fixable Aqua viability dye, anti-mouse CD11b PE-Cy7 (250×) (clone M1/70, 101215, BioLegend, San Diego, VT, USA), Ly6G PE-CF594 (100×) (clone 1A8, 562700, BD Biosciences, Erembodegem, Belgium), Ly6C PerCP-Cy5.5 (100×) (clone HK1.4, 128011, BioLegend, San Diego, VT, USA), MHC-II FITC (500×) (clone M5/114.15.2, 11-5321-82, Thermo Fisher Scientific, Waltham, USA), CD11c APCs (100×) (clone N418, 117309, BioLegend, San Diego, VT, USA), and XCR1 PE (100×) (clone ZET, 148204, BioLegend, San Diego, VT, USA). Neutrophils, monocytes, MHC-II^high^ macrophages, MHC-II^low^ macrophages, cDC1s, and cDC2s were sorted in the wells of a 96-well U-bottomed plate using a BD FACSymphony S6 sorter. Following the sorting, the cells were cultured for 24 h in RPMI-1640 + 10% FBS + penicillin/streptomycin at 37 °C, 5% CO_2_, after which the release of IL-1β was detected in the conditioned supernatant via ELISA (DY401-05, Bio-Techne, Minneapolis, MI, USA). For flow cytometry analysis of antiviral T cell immunity at an acute timepoint (7 days) and a memory timepoint (6 weeks post-infection), mice were given an i.v. injection of anti-mouse CD3 FITC (5 μg/mouse) (clone 17A2, 100203, BioLegend, San Diego, USA) in 200 μL of PBS 3 min prior to euthanasia. Fc blocking, cell fixation, and permeabilization were performed as described above. The cells were stained for 1 h at 4 °C with LIVE/DEAD Fixable Aqua viability dye, anti-mouse CD8 AF700 (250×) (clone 53-6.7, 100730, BioLegend, San Diego, VT, USA), IAV NP H-2Db (ASNENMETM), pentamer PE (10 μL/test) (WP/8302-06, ProImmune, Oxford, UK), CD44 PerCP-Cy5.5, CD62L APC-Cy7, CD69 APC-Cy7 (100×) (clone H1.2F3, 104525, BioLegend, San Diego, VT, USA), CD103 PerC-Cy5.5 (100×) (clone 2E7, 121415, BioLegend, San Diego, VT, USA), Bcl-2 AF647 (100×) (clone BCL/10C4, 633509, BioLegend, San Diego, VT, USA), Eomes PE-CF594 (100×) (clone X4-83, 567167, BD Biosciences, Erembodegem, Belgium), and T-bet PE-Cy7. For peptide restimulation (6 weeks post-infection), 1/5th of the lung and spleen single-cell suspensions were plated in RPMI-1640 + 10% FBS + penicillin/streptomycin in 96-well U-bottomed plates and stimulated for 6 h with anti-mouse CD28 (2 μg/mL) (clone 37.51, 14-0281-82, Thermo Fisher Scientific, Waltham, MA, USA) and a pool of three IAV NP MHC-I H-2Db peptides (ASNENMETM, RLIQNSLTI and GERQNATEI) (10 μM) (Genscript) in the presence of a Golgi plug (1000×) (BDB555029, Thermo Fisher Scientific, Waltham, MA, USA). Fc blocking, cell fixation, and permeabilization were performed as described above. The cells were stained for 1 h at 4 °C with LIVE/DEAD Fixable Aqua viability dye, anti-mouse CD8 AF700, IFN-γ APCs, and TNF PE (100×) (clone MP6-XT22, 506306, BioLegend, San Diego, VT, USA). The samples were recorded on a four-laser BD FACSymphony A3 flow cytometer (BD Biosciences, Erembodegem, Belgium). Data were analyzed using FlowJo software.

### 4.9. In Vitro Human moDC-CD8 T cell Co-Culture Experiments

PBMCs were thawed in warm complete RPMI-1640. Monocytes were purified by positive selection using CD14 MicroBeads (130-050-201 Miltenyi Biotech, Bergisch Gladbach, Germany), following the manufacturer’s instructions, and washed twice with complete RPMI-1640. The cells were seeded at a concentration of 1 × 10^6^ cells/mL in 6-well plates, stimulated with human IL-4 (250 U/mL) (574008, BioLegend, San Diego, VT, USA) and human GM-CSF (1000 U/mL) (572902, BioLegend, San Diego, VT, USA), and incubated at 37 °C for 5 days. On day 5, immature moDCs were collected and matured with 100 ng/mL LPS (tlrl-eklps, Invivogen, Toulouse, France) for 2 h. Half of the cells were pulsed with CMV peptides (100-0668, STEMCELL Technologies, Saint Égrève, France) for an additional 2 h, while half of the cells were left unpulsed. Another vial of PBMCs was thawed in warm complete RPMI-1640, after which negative CD8 selection was performed using a human CD8 T cell Isolation Kit (130-096-495, Miltenyi Biotech, Bergisch Gladbach, Germany). The isolated CD8 T cells were washed twice and stimulated with 0.5 nM WT IL-1β or vehicle control (PBS). Both moDCs and CD8 T cells were washed and co-cultured at a 1:1 ratio in a U-bottomed 96-well plate. After 20 h of co-culture, brefeldin A (1000×) was added for 4 h. The cells were washed and stained with anti-human CD45RO-BV421, CD4 BV605, CD27-PerCp-Cy7 (clone O323, 302838, BioLegend, San Diego, VT, USA), CD8-AF700, and Fixable Viability Dye eFluor™ 780 in a total volume of 50 µL for 30 min at 4 °C. After washing, the cells were fixed with the Foxp3 Transcription Factor Staining Buffer Set for 30 min and subsequently permeabilized according to the manufacturer’s instructions. Finally, the cells were stained with anti-human IFN-γ APC (clone 25723.11, 341117, BD Biosciences, Erembodegem, Belgium) for 30 min at 4 °C, washed, and resuspended in PBS. The samples were recorded on a three-laser BD LSRFortessa flow cytometer (BD Biosciences, Erembodegem, Belgium). Data were analyzed using FlowJo software.

### 4.10. Statistical Analyses and Data Presentation

Statistical analyses were performed using GraphPad Prism 9 software (GraphPad Software). Data are presented as the mean ± SEM in all experiments. Statistical testing was performed with an appropriate test, as described in the corresponding legends. Statistical significance was defined throughout as *p* < 0.05, and exact *p*-values are reported for all tests performed within the context of this work.

## Figures and Tables

**Figure 1 ijms-24-15770-f001:**
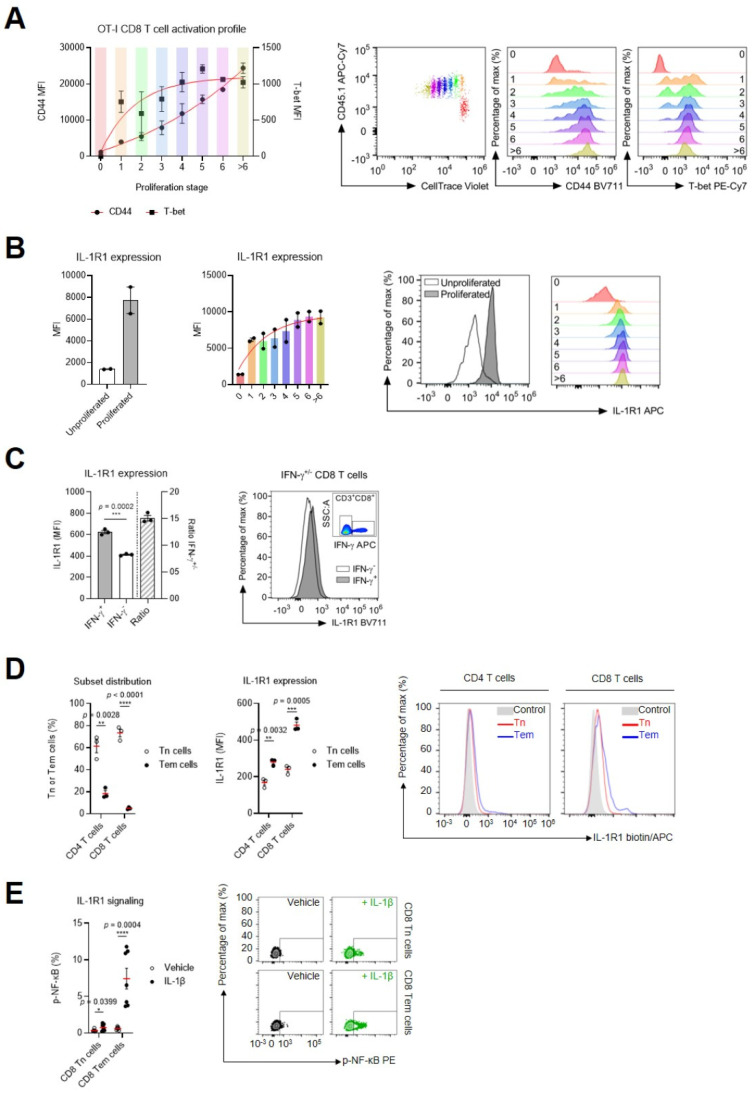
CD8 T cells dynamically modulate IL-1R1 levels during priming and maintain elevated sensitivity to IL-1 as memory cells: (**A**) CD44 (filled dots) and T-bet expression (filled boxes) in OT-I CD8 T cells over the course of their proliferation. The data are supported by representative flow cytometry plots. (**B**) Expression of IL-1R1 in unproliferated and proliferated OT-I CD8 T cells (left) and over the course of T cell proliferation (middle). The data are supported by a representative flow cytometry plot. Data points (**A**) or bars (**B**) represent the mean ± SEM of a representative set of three independent experiments. (**C**) IL-1R1 expression on IFN-γ^+/−^ CD8 T cells after ex vivo restimulation with SIINFEKL (100 nM). Representative flow cytometry plots are shown. Bars represent the mean ± SEM of an experiment with *n* = 3 mice/group. (**D**) Relative amounts of CD4 and CD8 Tn cells and Tem cells (based on the expression of CD44 and CD62L) isolated from the spleens of C57BL/6 mice (left). Surface expression of IL-1R1 within CD4 and CD8 Tn cells and Tem cell subsets (middle). Graphs are supplemented with representative flow cytometry plots, wherein the FMO signal is used as a control. The mean ± SEM of an experiment with *n* = 3 mice/group is shown. (**E**) Frequency of p-NF-κB^+^ cells within the CD8 Tn cell and CD8 Tem cell populations after stimulation of PBMCs for 15 min with IL-1β (0.5 nM). The mean ± SEM of an experiment with *n* = 7 human donors is shown. A representative flow cytometry plot is shown. ****, *p* < 0.0001; ***, *p* < 0.001; **, *p* < 0.01; *, *p* < 0.05; by unpaired Student’s *t*-test. See also Appendix A.

**Figure 2 ijms-24-15770-f002:**
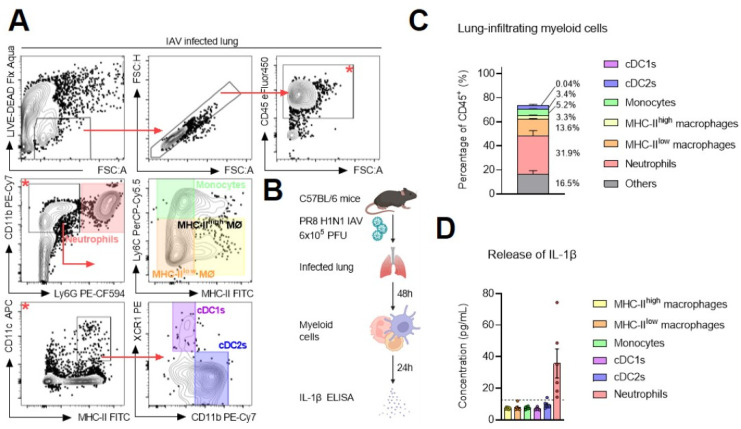
Neutrophils are the predominant source of IL-1β in the lungs during influenza A virus infection: (**A**) Gating strategy (* refers to the parent gate) and (**B**) experimental scheme for the sorts of different myeloid cell populations from lungs of PR8 IAV-infected C57BL/6 mice (6 × 10^5^ PFU intranasally) 48 h post-inoculum. (**C**) Proportions of myeloid cell subsets within the total leukocyte population in the lungs of PR8 IAV-infected C57BL/6 mice 48 h post-inoculum. Bars represent the mean ± SEM of an experiment with *n* = 6 mice/group. (**D**) Release of IL-1β (pg/mL) in the conditioned supernatants of sorted myeloid cell subsets 24 h post-sorting. See also Appendix A.

**Figure 3 ijms-24-15770-f003:**
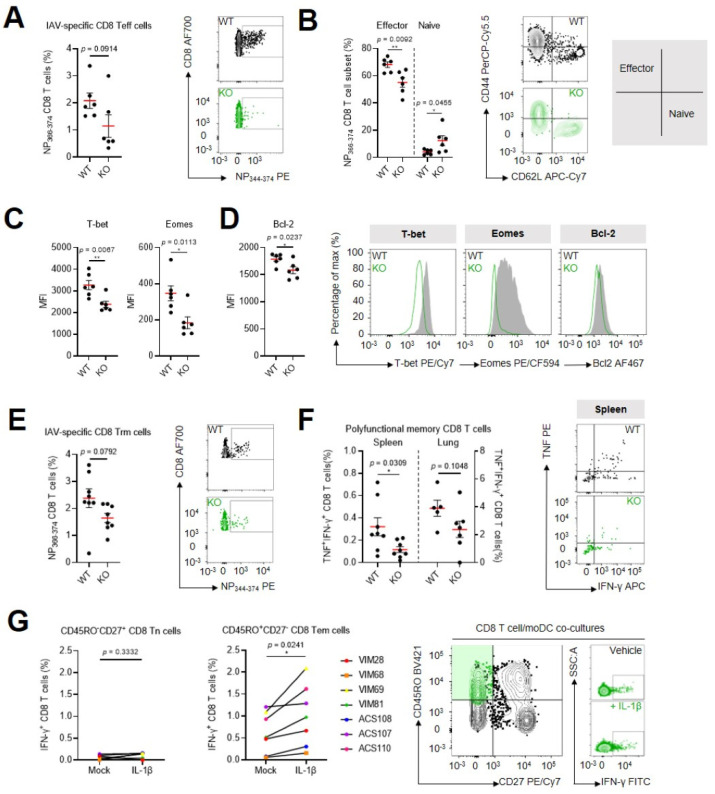
Loss of IL-1R1 signaling during IAV infection leads to impaired formation and functionality of antiviral CD8 Teff and CD8 Trm cells: (**A**) Frequencies of antiviral CD8 Teff cells within the total extravascular CD8 T cell population in lungs of WT and KO mice one week after PR8 H1N1 IAV infection (0.1LD50). (**B**) Frequencies of antiviral CD8 Teff cells that adopt an effector and a naive phenotype based on surface expression of CD44 and CD62L. (**C**,**D**) Expression of T-bet, Eomes (**C**) and Bcl-2 (**D**) in antiviral CD8 Teff cells. (**E**) Frequencies of antiviral CD8 Trm cells within the total extravascular CD8 Trm cell population in the lungs of WT and KO mice six weeks after PR8 IAV infection. (**F**) Frequencies of polyfunctional memory CD8 T cells that produce both IFN-γ and TNF after restimulation with PR8 IAV NP peptide pools within the total extravascular CD8 T cell population in the spleens and lungs of WT and KO mice six weeks after PR8 IAV infection. The mean ± SEM of an experiment with *n* = 6 mice/group (**A**–**D**) or a pool of two independent experiments with *n* = 7–8 mice/group combined (**E**,**F**) is shown. All graphs are supplemented with representative flow cytometry plots. **, *p* < 0.01; *, *p* < 0.05; by unpaired Student’s *t*-test. (**G**) Frequencies of IFN-γ^+^ CD8 Tn cells and CD8 Tem cells determined after 24 h of co-culture with autologous moDCs that present CMV peptide pools. CD8 T cells were left unstimulated or were stimulated for 1 h with 0.5 nM IL-1β co-culture initiation. Shown is a pool of two independent experiments with *n* = 7 human donors combined. Representative flow cytometry plots are included. *, *p* < 0.05; by paired Student’s *t*-test. See also Appendix A.

## Data Availability

All data supporting the findings of this study are available within the article.

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
