# Peer review of "Rethinking IL-1 Antagonism in Respiratory Viral Infections: A Role for IL-1 Signaling in the Development of Antiviral T Cell Immunity"

_ijms, 2023, doi:10.3390/ijms242115770_

Round 1

Reviewer 1 Report

Comments and Suggestions for Authors

The manuscript “Rethinking IL-1 antagonism in respiratory viral infections: A role for IL-1 signaling in the development of antiviral T cell immunity” by Bram Van Den Eeckhout with co-authors demonstrates the role of IL-1R in the development of specific CD8 immune response. Specifically, they demonstrated the dynamic upregulation of IL-1R levels during CD8 T cells interaction with APCs. IL-1R1 upregulation correlates with the proliferation index and effector status of newly activated CD8 T cells. IL-1 sensing persists in memory cells. Moreover, authors demonstrate impaired CD8 T cells response to influenza A virus in the absence of IL1 signaling.

Overall, the manuscript is timely, innovative, and of interest for the readers, but requires clarification of figures and discussion improvement.

Major comments

1. Figure 1 is very confusing and requires re-organization and more detailed legends. Fig 1 A. What is shown as circles and what is shown as squares? Place all pictures for fig 1A organized from left to right and from top to bottom or give subletters like Aa, Ab etc. or use boxes to combine all pictures for 1A and 1B. It is difficult to find pictures based on the figure legends: “(a) CD44 and T-bet expression in OT-I CD8 T cells over the course of their proliferation (left). Expression of IL-1R1 in proliferated and proliferated OT-I CD8 T cells (middle) and over the course of T cell proliferation (right). Data points (left and right) or bars (middle) represent the mean ± SEM”. Please clarify Figure and legends.

2. Figure 2. Give an additional letter to (Fig. 2B) the diagram of Influenza A virus infection and describe it in the legends.  

3. Fig. 3. Label a quadrant that is used for quantification on the flow cytometry results for Fig. 3B.

4. Line 300. Methods. Mice. Provide a short description of transgenic and knockout mice regarding immune status.

5. Figure 2 provides a description of lung-infiltrating myeloid cells 48 h after influenza A infection and demonstrates that neutrophils are major cells expressing IL-1 beta. The composition of lung-infiltrating myeloid cells is dynamic. Neutrophil levels are significantly reduced 5-7 days after infection, at the time of specific CD8 cells activation and proliferation. Thus, other cells may contribute to the IL-1 beta levels.

6.  Airway epithelial and endothelial cells produce IL-1 beta during infection. They may be a source of IL-1beta during CD8 priming. Discuss this possibility.

7. The levels of specific CD8 effector cells, polyfunctional memory cells and activation are lower in the IL1r1-/- KO mice. Is IL-1R knocked out in the myeloid cells of this mice? The lower levels of APCs activation (DC and macrophages) may significantly reduce production of specific CD8 T effector and memory cells.

Minor comments

1. Line 24 Correct “of both mouse and men” to “mouse and humans”

2. Line 70 What are OT-I CD8 cells?

3. Line 86 Correct “of both mouse and men” to “mouse and humans”

4. Line 116 “is depicted” change to “is shown”

Comments on the Quality of English Language

The paper is clearly written and required only minor corrections. 

Author Response

First of all, we would like to thank the reviewer for his/her careful examination of our work. Please find a point-by-point response below:

Major comments

1. Figure 1 is very confusing and requires re-organization and more detailed legends. Fig 1 A. What is shown as circles and what is shown as squares? Place all pictures for fig 1A organized from left to right and from top to bottom or give subletters like Aa, Ab etc. or use boxes to combine all pictures for 1A and 1B. It is difficult to find pictures based on the figure legends:“(a) CD44 and T-bet expression in OT-I CD8 T cells over the course of their proliferation (left). Expression of IL-1R1 in proliferated and proliferated OT-I CD8 T cells (middle) and over the course of T cell proliferation (right). Data points (left and right) or bars (middle) represent the mean ± SEM”. Please clarify Figure and legends.

Author reply: We apologize for the confusion that this Figure might have caused. As mentioned in the Figure legend, CD44 expression within CD8 T cells in a certain stage of cell division has been depicted as circles, whereas T-bet expression has been shown as squares. We clarified this in the guiding legend text. In addition, we reorganized the Figure according to suggestions of the reviewer. This novel Figure has been included in the manuscript text.

2. Figure 2. Give an additional letter to (Fig. 2B) the diagram of Influenza A virus infection and describe it in the legends.

Author reply: This has been included in a novel version of Figure 2, which can be retrieved in the manuscript text.

3. Fig. 3. Label a quadrant that is used for quantification on the flow cytometry results for Fig. 3B.

Author reply: This has been included in a novel version of Figure 3, which can be retrieved in the manuscript text.

4. Line 300. Methods. Mice. Provide a short description of transgenic and knockout mice regarding immune status.

Author reply: We introduced the following part: "As a consequence of the Rag2-/-, OT-I mice have no functional T and B cells besides the T cells that carry the transgenic TCR. Full-body Il1r1 knockout causes defective responses to certain inflammatory agents and altered responses to various pathogenic organisms."

5. Figure 2 provides a description of lung-infiltrating myeloid cells 48 h after influenza A infection and demonstrates that neutrophils are major cells expressing IL-1 beta. The composition of lung-infiltrating myeloid cells is dynamic. Neutrophil levels are significantly reduced 5-7 days after infection, at the time of specific CD8 cells activation and proliferation. Thus, other cells may contribute to the IL-1 beta levels.

Author reply: We agree to the reviewer's comment and introduced the following part: "While neutrophils are clearly the main source of IL-1β within the myeloid compartment early in the infection (48h post-inoculation), other cells might take over this role when the amount of infiltrating neutrophils starts to wane over time."

Reference Ueki, H.; Wang, I.H.; Fukuyama, S.; Katsura, H.; da Silva Lopes, T.J.; Neumann, G.; Kawaoka, Y. In vivo imaging of the pathophysiological changes and neutrophil dynamics in influenza virus-infected mouse lungs. PNAS 2018, 115, E6622-E6629 was introduced.

6.  Airway epithelial and endothelial cells produce IL-1 beta during infection. They may be a source of IL-1beta during CD8 priming. Discuss this possibility.

Author reply: The source of IL-1b during CD8 T cell priming by APCs can be highly variable, as mentioned by the reviewer. Different groups have shown that conventional dendritic cells (cDCs) (Jain A et al. Nat. Commun 2018) or specialized subsets of "hyperactive" type 1 conventional dendritic cells (cDC1s) (Zhivaki D et al. Cell Reports 2020 and Zanoni I et al. Science 2016) are capable of producing IL-1 in the context of priming. We therefore wish to refrain from discussing the exact cellular source of IL-1b production during adaptive immune cell priming in the context of this manuscript.

7. The levels of specific CD8 effector cells, polyfunctional memory cells and activation are lower in the IL1r1-/- KO mice. Is IL-1R knocked out in the myeloid cells of this mice? The lower levels of APCs activation (DC and macrophages) may significantly reduce production of specific CD8 T effector and memory cells.

Author reply: All cells have been targeted by this full-body knockout of the Il1r1 gene, including cells of myeloid origin. It is therefore very possible that the effects observed in this model are (to certain extents) mediated by IL-1 effects mediated via the innate immune system, rather than direct IL-1 activity on CD8 T cells. We want to stress this by adding the following clarification to the discussion: "A limitation of our experimental model is the use of full-body Il1r1-/- animals, from which we cannot deduct whether experimental observations following infection are necessarily driven by direct IL-1 activity on CD8 T cells."

Minor comments

1. Line 24 Correct “of both mouse and men” to “mouse and humans”

2. Line 70 What are OT-I CD8 cells?

3. Line 86 Correct “of both mouse and men” to “mouse and humans”

4. Line 116 “is depicted” change to “is shown”

Author reply: Minor comments were highlighted in red in the updated version of the manuscript.

Reviewer 2 Report

Comments and Suggestions for Authors

I congratulate the authors for conducting such a well-designed and executed study. It reads well and is essential for understanding the role of IL-1β in the anti-viral CD8 T-cell immune response to IAV. I have included minor comments in the attached PDF. Please find it attached.

Best regards,

Author Response

First of all, we would like to thank the reviewer for his/her careful examination of our work. Please find a point-by-point response below:

Minor comments

1. Could authors please list those cell population from bottom to up as shown in fig B? So freq and the respective cell populations can be aligned and easy to see. 

Author reply: This is a good suggestion by the reviewer. A new Figure 2 can be retrieved in the main text.

2. Can authors please increase the size of B and C to fill the empty spaces? So, the figures and titles can be more visible to eye. 

Author reply: This is a good suggestion by the reviewer. A new Figure 2 can be retrieved in the main text.

3. How authors defined "dominated by neutrophils" ?I assume authors observed these domination within CD45 expressing cells compared to other CD45+ cells. Is that correct? Please clarify. 

Author reply: The reviewer is correct. We clarified that we are looking within the subset of CD45+ cells in the main text: "FACS analysis of the total leukocyte population (all CD45+ cells) revealed that the myeloid compartment in IAV-infected lung parenchyma is dominated by neutrophils (mean = 31.88%; SEM = 4.38%) (Fig. 2A – C). MHC-IIlow macrophages (mean = 13.6%; SEM = 1.86%), monocytes (mean = 5.22%; SEM = 0.37%), cDC2s (mean = 3.4%; SEM = 0.55%) and MHC-IIhigh macrophages (mean = 3.27%; SEM = 0.33%) make up much smaller fractions of the myeloid repertoire."

4. Was it expected to see cDC1 at that time point? If not, why authors failed? 

Author reply: We rephrased this sentence to clarify that this is not a technical error. As supported by the references included, others have demonstrated cDC1 migration to the draining LN in the acute phase of IAV infection, which can explain their absence in the lung parenchyma.

5. It`s not necessary, but I wonder does authors measured the pro-IL1b too? If so, the ratio of pro IL-1b over the bio-active IL-1b can be supplemented as a figure.

Author reply: While interesting, we did not conduct these analyses.

6. I assume authors can show this by gating IL-1b in the neutrophil cells. Is that possible? 

Author reply: This is technically possible, but in this experiment, we sorted neutrophils from the infected lungs and measured release of IL-1b by the sorted cells in the conditioned supernatant with ELISA.

7. I see, but what is this mean? Why is it important?

Author reply: The reference included backs up our experimental data, and we believe that this could represent a possible mechanistical observation of how IL-1 signaling in T cells potentially drives Eomes expression.

8. There is a direct depletion of CD8 T cell methods such as use of anti-CD8 mab, perhaps antigen specific CD8 T cell depletion methods. Why this information should replace the current approaches?

Author reply: This is not the main message of our work. With our data, we want to raise the point that the consequences of IL-1 inhibition (which is currently pursued in clinical applications) on the rollout of CD8 T cell immunity should be more carefully considered and taken into account. Secondly, we believe that our new data supports the importance of IL-1 activity in the mode of action of T cell vaccines and immunotherapies aimed at the generation of a CD8 T cell response.

Round 2

Reviewer 1 Report

Comments and Suggestions for Authors

Authors addressed all my comments. 

Comments on the Quality of English Language

Minor editing of English language required.